# OpenReview forum: "Intersectional Stereotypes in Large Language Models: Dataset and Analysis"
_EMNLP/2023/Conference — EMNLP 2023 Findings_

### Official Review · Reviewer_k5qo · 2023-07-21

**Typos Grammar Style And Presentation Improvements:** 1. Inconsistent writing
**Soundness:** 3

**Ethical Concerns:**

Yes

**Excitement:**

3: Ambivalent: It has merits (e.g., it reports state-of-the-art results, the idea is nice), but there are key weaknesses (e.g., it describes incremental work), and it can significantly benefit from another round of revision. However, I won't object to accepting it if my co-reviewers champion it.

**Justification For Ethical Concerns:**

No details on the human study: IRB, how the annotators were recruited, demographics of annotators (this is important for stereotypes studies as if the annotators come from the out-groups they might know certain stereotypes or at least we need to know how diverse was the crowd), how did they task look like exactly?

**Paper Topic And Main Contributions:**

The authors propose a dataset of intersectional stereotypes that would account for several identities (7 cases for 4 identities, half of the dataset for two identities, and the rest for three identities).

The authors consider  6 categories: race, age, gender, religion, political preferences, and disability status. Stereotypes are generated by ChatGPT, then filtered by ChatGPT and humans, and then human validated. in total, there are 106 groups with at least 1 stereotype per group and a maximum of 11 stereotypes, the average is 4.53

**Questions For The Authors:**

1. I don't see how the statement of strengths of Chatgpt follows the sentence about its problem with overgeneralization. Why does incorporating various mitigation techniques to fight the model's overgeneralization show the strength of the model? What do I miss?
2. Could you please elaborate more on life roles? I would love to see more on why exactly you need them - to confirm stereotypes or what? How they were verified? What is the number of these stories? Are they in the dataset? What are the labels related to these stories?
3. If you are saying that you manually investigated the generations: how many of them there were in total?

**Reasons To Accept:**

Dataset of intersectional stereotypes for a wide variety of groups.

**Reasons To Reject:**

Looped data collection and non-transparent process of data-quality verification.

ChatGPT generates stereotypes for the intersectional identities, then verifies them, them human annotators (no survey details or overall details on how they verify the data quality was provided, how many questions every annotator did?).

Stereotypes are extremely subjective and it is important to know who verified and said that this is the stereotype about this very particular social group, given that we have several identities included.

Unclear format of the dataset: what is included? Having a few examples would be nice.

I would also expect some discussion on stereotypes(except angry black women): what are they, and do they relate to the existing research (in any field)?



**Reproducibility:**

1: Could not reproduce the results here no matter how hard they tried.

**Reviewer Confidence:**

5: Positive that my evaluation is correct. I read the paper very carefully and I am very familiar with related work.

---

> ### Author Rebuttal · Authors · 2023-08-28
>
> Thank you for your thoughtful review of our manuscript. We genuinely appreciate the time you took to provide detailed feedback. We acknowledge the areas you identified for improvement and below, we address each of your concerns systematically. We are open to further discussions or clarifications during the reviewer-author phase.
>
> **Data collection and validation process**: We understand the confusion our initial description might have caused. Here's a more organized breakdown:
>
> * We started by constructing demographic groups based on established stereotype research.
>
> * Next, intersectional demographic groups were created by merging various demographic features.
>
> * ChatGPT generated stereotypes for each group, which were then manually refined with assistance from the model to sift out unlikely stereotypes.
>
> * Finally, we employed human validation to match these stereotypes with commonly held beliefs, to ensure that the recorded stereotypes align with general awareness.
>
> Our paper's Section 2 offers an outline of this process. Due to space constraints, specific questionnaires for human validation were excluded but will be elaborated on in our revised version. It's essential to note that each annotator addressed two key questions, allowing for a detailed examination of the stereotype's prevalence and its potential application to broader demographic groups.
>
> **Annotator Background**: We agree that stereotype perception is inherently subjective. Given the nature of our study, which revolved around the recognition and understanding of stereotypes, we were chiefly concerned with general awareness and knowledge rather than specific personal experiences or demographic backgrounds. For this reason, we did not actively document the individual backgrounds of our annotators. We wanted their responses to reflect widespread perceptions, acting as a proxy for societal awareness of specific stereotypes.
>
> As for training, we intentionally refrained from providing extensive training to the annotators. The rationale behind this decision was rooted in our desire to capture the genuine, unfiltered, and spontaneous reactions of individuals when confronted with certain stereotypes. Introducing formal training could, in effect, have guided or biased their responses. Our expectation was for annotators to navigate our questionnaires based purely on their personal knowledge and understanding of stereotypes.
>
> We also took a deliberate stance on not rigidly controlling the background of our annotators. Our emphasis was to gauge the pervasiveness of a stereotype in the broader population rather than its impact or recognition within particular demographic groups.
>
> Our post-analysis of the data reinforced our methodology, as we observed a commendable level of consistency in the responses from our annotators. This consistency bolstered our confidence in the approach, suggesting that our annotators, despite their varied backgrounds, shared common perceptions about the stereotypes under investigation.
>
> **Dataset Format Clarification**: We appreciate the suggestion to include concrete examples from the dataset. In the updated version of our paper, this will be addressed for clarity. Our dataset provides statistics on stereotypes targeting 106 intersectional groups, as outlined in Table A1. It also details the content and explanations of these stereotypes, with a representative example showcased in Lines 143-145. Additionally, the dataset contains questions, along with their anticipated answers, which were utilized to assess the stereotypes in large language models, as presented in Table B2. Should our paper be accepted, both the dataset and the code used for its generation will be made publicly accessible.
>
> **Discussion of stereotypes**: We appreciate your insight on contextualizing stereotypes within existing research. Space limitations initially restricted extensive discussions, especially since our identified stereotypes align with existing research. We will expand on this, including more examples in our revised edition. For instance, our dataset points towards a stereotype about Asian women being perceived as "obedient and submissive". This particular stereotype is not a novel observation, as it has been discussed and documented in the literature [1]. Similarly, another stereotype we identified pertains to Black men, who are sometimes unfairly characterized as "absent fathers" and perceived as lacking in family values. This too echoes findings in prior studies [2].
>
> What makes our dataset particularly valuable, we believe, is its exploration of nuanced intersectional groups. For example, it touches upon perceptions surrounding "Black young men", diving deeper than broad demographic categories. While these intricate intersectional stereotypes are less frequently discussed in conventional studies, their inclusion in our dataset provides a richer and more detailed tapestry of societal perceptions.
>
> In our revised paper, we will incorporate a more detailed discussion, highlighting the significance of these stereotypes, their origins, and their implications in the context of our research.
>
> **Strengths of ChatGPT in stereotype research**: We regret the lack of clarity here. Our intention was to emphasize ChatGPT's utility in refining our raw dataset, especially with well-curated, close-ended questions, streamlining the data filtering process. However, the model did exhibit overgeneralization tendencies when initially forming the raw dataset using open-ended questions.
>
> **Role of life experiences in testing stereotypes**: Your suggestion regarding the potential pitfalls of using static prompts for evaluating the presence of stereotypes in LLMs is well taken. We believe that dynamic, rich contexts offered by virtual life experiences can better reveal the ingrained biases in these models. The use of life roles or virtual life experiences is crucial to our methodology for several reasons:
>
> * **Avoiding Monotonous Answers**: If we prompt LLMs using the same, unchanging description of a target group, like "suppose you are a black male high school student, how did you do in your math exam last week?", it may lead to consistent, unchanged answers from the model. In such a scenario, it becomes challenging to differentiate between the model's inherent stereotyping and its consistency due to repeated prompts.
>
> * **Dynamic Contextual Testing**: By having the LLM adopt different life roles, each with unique demographic or psychographic features, while retaining the same controlled intersectional group, we introduce variability. If the model's responses to a consistent question still show homogeneity, even with varied life stories, it becomes clearer that the model harbors certain stereotypes.
>
> * **Diversifying Uncontrolled Characteristics**: The generated virtual life experiences allow us to change the uncontrolled characteristics of the individuals being described. By doing this, we keep the controlled demographic features consistent, which provides a stable foundation for testing while introducing the necessary variability to assess the model's behavior.
>
> * **Validation of Life Roles**: It's essential to clarify that we did not randomly generate these life roles. As noted in Line 273 of our paper, we painstakingly created 10 unique life roles for each intersectional group. These were meticulously reviewed by native English speakers from our team. This review ensured that the language used was fluent and coherent and that the stories provided diverse, non-repetitive insights into the intersectional group in question.
>
> * **Transparency and Openness**: We're committed to transparency in our research. Hence, each life role, labeled only by its controlled demographic group, will be made publicly available alongside our dataset upon the acceptance of our paper.
>
> By integrating these life roles into our research process, we can provide a more accurate, nuanced, and comprehensive evaluation of how LLMs, such as ChatGPT, respond to varying intersectional groups and the potential stereotypes they might perpetuate.
>
> **Number of stereotypes**: Table A1 of Appendix A provides details of the exact distribution of stereotypes across different intersectional groups. This table not only gives an overview of the breadth of our dataset but also underscores the granularity with which we've addressed stereotypes. Additionally, we avoided repetitions in two main ways: (i) Within-group Repetition: The first challenge was ensuring that there weren't repeated stereotypes within the same intersectional group. Given the breadth of stereotypes and their subtle variations, manual oversight was crucial. We made sure that each stereotype attributed to a particular intersectional group was unique. (ii) General vs. Specific Groups: Another layer of complexity arose when considering general demographic groups relative to more specific ones. For example, the group "male" is more generalized compared to "black male," but "female" stands independently. Our approach ensured that we did not redundantly assign stereotypes to both a general group and its more specific counterpart. This ensures that the stereotypes in our dataset are not just broad strokes but also delve into the nuances of intersectionality.
>
> While avoiding repetition, we also maintained a careful balance to ensure that we did not exclude stereotypes that might be relevant to both a broader demographic as well as its sub-demographics. This was a delicate balancing act, ensuring that our dataset was both exhaustive and free from redundancy. By following these steps, we've aimed to curate a dataset that captures the nuances of intersectional stereotypes without falling into the pitfalls of redundancy or oversimplification.
>
> **Writing inconsistencies and clarifications**: To clarify, we did not consider appearances and illegal behaviors as bad indicators of stereotypes. Instead, we think that to fully grasp stereotypes, one must look beyond just physical appearances or illicit behaviors. Stereotypes are nuanced societal perceptions encompassing various attributes, attitudes, and behaviors. In our research, we delved into the multifaceted nature of stereotypes instead of taking a limited view. We avoided arbitrary boundaries in our data collection. We formulated questions not just on appearances or illegal behaviors, but on perceptual beliefs and ingrained attitudes, such as societal responses to authority. We will make this clearer in our revised paper.
>
> **Mentions of Demographic Features Confusing**: Thank you to the reviewer for highlighting this. We recognize that the interchangeable use of the terms “feature” and “category” may lead to confusion. We will rectify this in the revised version of the paper to ensure clarity.
>
>
>
> [1] Uchida, Aki. "The orientalization of Asian women in America." Women's Studies International Forum. Vol. 21. No. 2. Pergamon, 1998.
>
> [2] Smith, C. A., Krohn, M. D., Chu, R., & Best, O. (2005). African American Fathers: Myths and Realities About Their Involvement With Their Firstborn Children. Journal of Family Issues, 26(7), 975–1001. https://doi.org/10.1177/0192513X05275421

---

### Official Review · Reviewer_PMWn · 2023-07-24

**Soundness:** 3

**Excitement:**

3: Ambivalent: It has merits (e.g., it reports state-of-the-art results, the idea is nice), but there are key weaknesses (e.g., it describes incremental work), and it can significantly benefit from another round of revision. However, I won't object to accepting it if my co-reviewers champion it.

**Paper Topic And Main Contributions:**

The paper introduces an intersectional stereotype dataset and evaluates the prevalence of stereotypes in LLMs across 106 intersectional groups. The stereotypes in the dataset were classified into 16 categories. The authors also provide a comprehensive analysis of intersectional stereotype propagation in different LLMs by leveraging this dataset.

**Questions For The Authors:**

Q1: If this paper is accepted, will the dataset and the code be publically available?
Q2: What are the backgrounds for the annotators? How much does it cost in terms of time? Did you train them?

**Reasons To Accept:**

- well-written paper
- new dataset
- sound experiments
- interesting problem

**Reasons To Reject:**

- The guidelines given to the annotators were not introduced or discussed. Also, no information about the annotators was included (i.e. their backgrounds).

**Reproducibility:**

3: Could reproduce the results with some difficulty. The settings of parameters are underspecified or subjectively determined; the training/evaluation data are not widely available.

**Reviewer Confidence:**

3: Pretty sure, but there's a chance I missed something. Although I have a good feel for this area in general, I did not carefully check the paper's details, e.g., the math, experimental design, or novelty.

**Typos Grammar Style And Presentation Improvements:**

Line 267: the reference (Argyle et al.,2022) should come before the full stop.
Line 291: extra space "3.  We"

---

> ### Author Rebuttal · Authors · 2023-08-28
>
> We would like to express our sincere gratitude for the time and effort you devoted to reviewing our paper and providing insightful feedback. We appreciate  your acknowledgment of the novelty of our dataset, the rigor of our experiments, the quality of our writing, and the significance of our topic. We acknowledge the areas you identified for improvement and below, we address each of your concerns systematically. We are open to further discussions or clarifications during the reviewer-author phase.
>
> **Annotation Guideline**: We apologize for the oversight in not detailing the specific questionnaire used for our manual validations. Your point is well-taken. Although a description is provided in Section 2.5, we agree that presenting an example questionnaire, inclusive of detailed instructions, would bolster clarity and understanding of our validation process. We commit to incorporating this in our revised manuscript.
>
> **Information about Annotators**:
>
>
>
> * The annotators were enlisted from Amazon Mechanical Turk. One key criterion we had in place was that these annotators should be residents of English-speaking countries. This decision was taken to ensure a certain level of linguistic and cultural familiarity, which we believed would be pivotal to the recognition of specific stereotypes in our dataset.
>
> * The primary task of the annotators during the human validation process was twofold: Firstly, to identify if they could recognize a specific stereotype directed towards a particular intersectional group. Secondly, to discern if such stereotypes were more commonly associated with a broader demographic group rather than the specific intersectional group in question.
>
> * Given the nature of these tasks, it was not imperative for the annotators to hail from the intersectional groups being examined. Our focus was more on the general awareness of these stereotypes rather than personal experiences tied to them. This is why we refrained from collecting detailed background information on our annotators. We felt that doing so might introduce biases or preconceived notions which could, inadvertently, skew the results of our study.
>
> * However, we acknowledge the importance of this concern and will include these details and our rationale behind such decisions more comprehensively in the revised version of our paper.
>
> **Annotator Background & Training**: Given the nature of our study, which revolved around the recognition and understanding of stereotypes, we were chiefly concerned with general awareness and knowledge rather than specific personal experiences or demographic backgrounds. For this reason, we did not actively document the individual backgrounds of our annotators. We wanted their responses to reflect widespread perceptions, acting as a proxy for societal awareness of specific stereotypes.
>
> As for training, we intentionally refrained from providing extensive training to the annotators. The rationale behind this decision was rooted in our desire to capture the genuine, unfiltered, and spontaneous reactions of individuals when confronted with certain stereotypes. Introducing formal training could, in effect, have guided or biased their responses. Our expectation was for annotators to navigate our questionnaires based purely on their personal knowledge and understanding of stereotypes.
>
> We also took a deliberate stance on not rigidly controlling the background of our annotators. Our emphasis was to gauge the pervasiveness of a stereotype in the broader population rather than its impact or recognition within particular demographic groups.
>
> Our post-analysis of the data reinforced our methodology, as we observed a commendable level of consistency in the responses from our annotators. This consistency bolstered our confidence in the approach, suggesting that our annotators, despite their varied backgrounds, shared common perceptions about the stereotypes under investigation.
> **Dataset & Code Availability**: We will make our dataset public. Alongside the dataset, our code, with comprehensive documentation, will also be made available. We believe this will be a valuable resource for the wider research community, especially in the domain of intersectional stereotypes.

---

### Official Review · Reviewer_kDnu · 2023-08-04

**Soundness:** 4

**Excitement:**

3: Ambivalent: It has merits (e.g., it reports state-of-the-art results, the idea is nice), but there are key weaknesses (e.g., it describes incremental work), and it can significantly benefit from another round of revision. However, I won't object to accepting it if my co-reviewers champion it.

**Missing References:**

The paper does not include references to shared tasks where the stereotype is investigated, nor the linguistic and psychological literature in which the notion of stereotype is defined.

**Paper Topic And Main Contributions:**

The paper introduces a novel dataset built for intersectional stereotypes and some experiments for investigating the prevailing stereotypes in different language models.
It is described the process for creating the dataset, which includes the automatic generation and the manual validation of stereotypical texts.
A set of questions is built for testing the behaviour of language models with respect to intersectional stereotypes.

**Questions For The Authors:**

In general, I do not understand why the authors did not choose to send a regular long paper instead of sending a short one in which 1) they had to reduce too much images, 2) they had to put in the appendix material that should have been in the text as the subject of useful discussion and comparison, and 3) they had to not include the necessary discussion of the reference literature (in a section of related work or elsewhere in the paper).

For example, a clear definition of what the authors mean by stereotype is necessary as starting point.

The investigation of stereotypes is among the recent trends in NLP and a few works have been published about their automatic detection, while in a few campaigns shared tasks have been organized for evaluating the performance in this task. They should be cited as a general reference before starting the discussion about their intersection.

In section 3.2 the sentence describing the first step of the analysis procedure is for me completely obscure.
The exploitation of some graphics may be of help in presenting the procedure and its steps.

The lack of examples makes it very hard to understand the meaning of some points.

**Reasons To Accept:**

The paper introduces an interesting topic and a novel resource for starting its investigation.

**Reasons To Reject:**

The methodology is not clear in all its details and the comparison with previous work is very limited. The presentation and discussion of some examples could have served as a clarification, but no example is proposed in the paper.

The literature cited is in practice limited to that related to language models and nothing is said about the (psychological and linguistic) notions on which this study should be found (stereotype and intersectionality), nor is any reference made to the evaluation tasks in which stereotypes were started to be addressed.

**Reproducibility:**

3: Could reproduce the results with some difficulty. The settings of parameters are underspecified or subjectively determined; the training/evaluation data are not widely available.

**Reviewer Confidence:**

3: Pretty sure, but there's a chance I missed something. Although I have a good feel for this area in general, I did not carefully check the paper's details, e.g., the math, experimental design, or novelty.

**Typos Grammar Style And Presentation Improvements:**

I've not detected typos in this paper.
However, I noticed that the figures (all containing text) are actually unreadable because of the microscopic size of the characters.

---

> ### Author Rebuttal · Authors · 2023-08-28
>
> Thank you for your comprehensive review and constructive feedback. We appreciate the effort and time you took in evaluating our work. We are also deeply appreciative of your positive comments regarding the novelty of our research topic and dataset. We hope our responses below sufficiently address your concerns, and we remain open to further discussions and suggestions during the reviewer-author discussion phase.
>
> **Previous Work**: We acknowledge the oversight in not discussing previous works on stereotypes and intersectionality in greater detail. While our main intention was to highlight the uniqueness of our dataset, especially given the absence of shared tasks specifically targeting intersectional stereotypes, we understand the importance of situating our work within broader psychological and socio-linguistic contexts. In the subsequent revision, we will incorporate foundational literature from these domains, such as Ghavami et al. [1] and Johnson et al. [2] and provide a more thorough discussion of the concepts.
>
> **Clarity and Presentation**: We apologize for the ambiguity arising from relegating important information to the appendix due to page constraints. Should our paper be accepted, we will prioritize relocating essential content to the main body (utilizing the additional page given to accepted papers). This should streamline the reader's experience, minimizing the need to refer to the appendix frequently.
>
> **Definition of Stereotypes**: We thank you for emphasizing the importance of a clear definition. For our research, we followed Blum's [3] definition of stereotypes as “ false or misleading generalizations about groups, generally widely shared in a society, and held in a manner resistant, but not totally, to counterevidence. Stereotypes shape the stereotyper’s perception of stereotyped groups, seeing the stereotypic characteristics when they are not present, and generally homogenizing the group.” We will ensure this definition is made more prominent in our revision.
>
> **Obscure Stereotype Examination Process**: Our method for investigating stereotypes in large language models follows a systematic approach, with the goal to discern patterns and tendencies in responses that could be classified as perpetuating stereotypes. Specifically:
>
> * Question Repository: As indicated in Line 253, our primary source of evaluation queries is stored in Table B2 in Appendix C. We acknowledge the oversight and regret the typographical error, and it should indeed have been Table C1.
>
> * Expected Responses: Accompanying these queries, we also provided anticipated or 'neutral' responses. These responses serve as a yardstick against which model responses can be gauged.
>
> * Simulation of Intersectional Roles: To dive deep into the potential intersectional biases of the models, we framed our evaluation in terms of roles that represent various intersectional groups in our dataset. Each role embodies an intersectional group associated with at least one stereotype.
>
> * Analysis of Response Distribution:
>
>     * For each question, we feed it to the language model, requesting it to answer from the perspective of the simulated roles.
>     * We then scrutinize the spectrum of answers produced by the model.
>     * Our primary metric of interest is the diversity in the responses: if a large portion of the model's answers tend to converge on a singular possible reply, it provides us a clue. A high convergence indicates the model is defaulting to a specific type or category of answer, which could be a manifestation of its inherent stereotypical bias.
>     * For instance, if 90% of the time, for a particular intersectional role, the model replies with a certain stereotype, it's indicative of the model's inclination to stereotype that role.
> * Sensitivity of Questions: Our queries were not chosen haphazardly. They have been meticulously crafted to be highly sensitive to the stereotypes in our dataset. Therefore, a biased response to these questions is not merely an inadvertent mistake but a more systematic reflection of the underlying biases.
> * Future Visualization Enhancements: We genuinely appreciate the suggestion to employ graphics for elucidating our stereotype examination process. A well-structured visual representation, such as a flowchart or process diagram, can significantly enhance comprehension, enabling readers to grasp the nuances more swiftly. We are committed to incorporating this in the revised paper.
>
> **Missing References**: We recognize the omission of some literature and shared tasks relating to stereotypes. We will expand our literature review section in the revision to cover these works.
>
> [1] Ghavami, Negin, and Letitia Anne Peplau. "An intersectional analysis of gender and ethnic stereotypes: Testing three hypotheses." Psychology of Women Quarterly 37.1 (2013): 113-127.
>
> [2] Johnson, Kerri L., Jonathan B. Freeman, and Kristin Pauker. "Race is gendered: how covarying phenotypes and stereotypes bias sex categorization." Journal of personality and social psychology 102.1 (2012): 116.
>
> [3] Blum, Lawrence. "Stereotypes and stereotyping: A moral analysis." Philosophical papers 33.3 (2004): 251-289.

---

### Meta-Review · Area_Chair_8TzQ · 2023-09-18

**Recommendation:** 3

**Metareview:**

Note: This paper involves human-subjects recruited from Amazon Mechanical Turk. Any kind of research that involves paid human-subjects requires an approved IRB. The human-tasks involved in the study required annotation of datasets including potentially harmful content. This further warrants that the proposed submission can only be accepted with an approved IRB. Without an IRB the paper should not be accepted. The current ethic statement or the authors' responses in the rebuttal do not address this issue.


Most reviewers find the soundness of this work to be good, providing sufficient support for its major arguments. All, are however, ambivalent in terms of excitement. Several key areas  of improvement highlighted by the reviewers can make the submission stronger as summarized below, to which the authors have sufficiently responded to in their revision plan.

1.	Improve overall clarity of presentation/ texts
2.	Clarify annotator guideline/ background/ survey details
3.	Define stereotype and provide more examples
4.	Discuss limitations of subjective interpretation around stereotypes (e.g., what one ethnic group might consider offensive is not to others, in other words, there is limitations around the generalizability of stereotypes the authors are discussing; how is this resolved in the annotation process?

Overall the authors address these concerns raised by the reviewers, however there is no discussion of an IRB for this submission, which is required for such studies that involve paid human subjects.

---

### Decision · Program_Chairs · 2023-10-07

**Decision:**

Accept-Findings

**Comment:**

Note: This paper involves human-subjects recruited from Amazon Mechanical Turk. Any kind of research that involves paid human-subjects requires an approved IRB. The human-tasks involved in the study required annotation of datasets including potentially harmful content. This further warrants that the proposed submission can only be accepted with an approved IRB. Without an IRB the paper should not be accepted. The current ethic statement or the authors' responses in the rebuttal do not address this issue.


Most reviewers find the soundness of this work to be good, providing sufficient support for its major arguments. All, are however, ambivalent in terms of excitement. Several key areas  of improvement highlighted by the reviewers can make the submission stronger as summarized below, to which the authors have sufficiently responded to in their revision plan.

1.	Improve overall clarity of presentation/ texts
2.	Clarify annotator guideline/ background/ survey details
3.	Define stereotype and provide more examples
4.	Discuss limitations of subjective interpretation around stereotypes (e.g., what one ethnic group might consider offensive is not to others, in other words, there is limitations around the generalizability of stereotypes the authors are discussing; how is this resolved in the annotation process?

Overall the authors address these concerns raised by the reviewers, however there is no discussion of an IRB for this submission, which is required for such studies that involve paid human subjects.